# Does the Gain of Total Neoadjuvant Therapy Outweigh the Harm in Rectal Cancer? Importance of the ATRESS (neoAdjuvant Therapy-RElated Shortening of Survival) Phenomenon: A Systematic Review

**DOI:** 10.3390/cancers15041016

**Published:** 2023-02-05

**Authors:** Joanna Socha, Krzysztof Bujko

**Affiliations:** 1Department of Radiotherapy, Military Institute of Medicine–National Research Institute, ul. Szaserów 128, 04-141 Warsaw, Poland; 2Department of Radiotherapy, Regional Oncology Centre, ul. Bialska 104/118, 42-200 Czestochowa, Poland; 3Department of Radiotherapy I, Maria Sklodowska-Curie National Research Institute of Oncology, ul. W.K. Roentgena 5, 02-781Warsaw, Poland

**Keywords:** total neoadjuvant therapy, rectal cancer, quality of life

## Abstract

**Simple Summary:**

Total neoadjuvant therapy (TNT) refers to delivering both (chemo)radiation and chemotherapy before surgery, aiming to reduce incidence of distant metastases. Indeed, in two randomised trials, TNT reduced distant metastases incidence compared with neoadjuvant chemoradiation, but without improvement of overall survival (OS), and at the expense of severe toxicity. We aimed to evaluate whether TNT results in long-term OS or quality of life (QoL) benefit. Systematic review of randomised trials comparing TNT with neoadjuvant (chemo)radiation identified six trials. Follow-ups were too short to resolve whether TNT improves long-term OS. The ATRESS phenomenon (i.e., reduction in post-metastatic survival) could explain a discrepancy between reduction of distant metastases and the absence of OS improvement after TNT in one trial. QoL analysis performed based on one trial’s data, showed that QoL might not have been improved with TNT if all patients are being considered for radical resection.

**Abstract:**

Background: We aimed to evaluate whether total neoadjuvant therapy (TNT) results in long-term overall survival (OS) or quality of life (QoL) benefit compared with chemoradiation if all patients are being considered for radical resection, and whether the ATRESS phenomenon (i.e., reduction in post-metastatic survival) impacts OS after TNT. Methods: Systematic review of randomised trials comparing TNT with neoadjuvant (chemo)radiation. Results: Six trials were identified. Follow-ups were too short to resolve whether TNT improves long-term OS. QoL analysis in one trial showed worse long-term neurotoxicity-related QoL (any neurotoxicity: 14% vs. 3%), higher rate of grade III+ acute toxicity (48% vs. 25%), longer duration of neoadjuvant treatment (19 vs. 6 weeks) and higher rate of locoregional failure (10% vs. 7%) in TNT vs. chemoradiation. This should be weighed against an absolute 8% reduction in the incidence of distant metastases (DM) after TNT. ATRESS could explain a discrepancy between reduction of DM and the absence of OS improvement after TNT in one trial. Conclusion: In the light of unproven OS benefit, the gain of TNT (reduction of DM) does not seem to outweigh the harm (excess of toxicity). ATRESS can be a reason for the absence of the OS benefit despite the reduction in DM.

## 1. Introduction

Total neoadjuvant therapy (TNT) in rectal cancer has been developed based on two premises: (1) the meta-analyses of randomised studies have not shown benefit in overall survival (OS) after adjuvant chemotherapy in patients who underwent neoadjuvant chemoradiation [1,2], while there is evidence showing better compliance to neoadjuvant than to adjuvant chemotherapy [3]; and (2) a hypothesis that effectiveness of chemotherapy in preventing distant metastases should be better if it is moved from post- to preoperative period. This is based on a logical assumption that subclinical distant disease before surgery is smaller, and therefore more chemosensitive, than in the postoperative period. It was, therefore, expected that after TNT, the incidence of distant metastasis would decrease, leading to an improvement in disease-free survival (DFS) and ultimately to an improvement in OS.

Indeed, the hypothesis of higher efficacy of TNT compared with chemoradiation in preventing distant metastases was confirmed by the two large randomised trials: RAPIDO by Bahadoer et al. [4] and PRODIGE-23 by Conroy et al. [5]. These trials met their primary endpoint of improvement in DFS (or in disease-related treatment failure) with TNT at 3 years compared with neoadjuvant chemoradiation; however, it was at the expense of severe toxicity. A benefit in OS with TNT was not demonstrated in either of these two trials.

The meta-analysis of randomised trials on adjuvant chemotherapy in stage III colorectal cancer showed that 3-year DFS benefit translated to OS benefit at 6–7 years [6]. However, no data supports such surrogacy in rectal cancer [7]. Moreover, RAPIDO, the largest randomised trial exploring TNT, demonstrated that the actuarial curves showing OS in the TNT cohort and in the chemoradiation cohort overlapped and there was no sign of curves diverging at 5 years [4]. Therefore, it is uncertain whether an OS benefit of TNT may occur with longer observation.

Generally, a benefit in DFS or in disease-related treatment failure after experimental toxic treatment without improving OS or quality of life (QoL) is not sufficient to change a standard of care. Thus, it is important to evaluate whether TNT has resulted in a long-term OS benefit or in QoL benefit. It is also important to assess the reasons for which the reduction of distant metastases did not translate into OS improvement with TNT [4,5]. One of the possible reasons could be shorter survival after the onset of distant metastases in the TNT group compared with the control group, as shown in the RAPIDO trial [8]. This phenomenon is termed ATRESS (neoAdjuvant Therapy-RElated Shortening of Survival) [9].

There are two main aims of this systematic review: (1) an evaluation of whether TNT results in a long-term OS benefit or QoL benefit; and (2) an assessment of the impact of ATRESS on OS after TNT. Additionally, we evaluated the importance of the ATRESS phenomenon after other neoadjuvant treatment schedules for rectal cancer because such assessment has not been performed previously.

Apart from an improvement of oncological outcomes, another purpose of TNT in rectal cancer is to improve QoL by organ preservation via watch-and-wait strategy (w&w) [10]. However, w&w is still not commonly implemented into routine practice. Therefore, in our review, we focus on TNT given with the intention of improving oncological outcomes, without considering its role in w&w.

## 2. Materials and Methods

### 2.1. Search Strategy

The systematic review was conducted as per the PRISMA guidelines [11]. The PubMed database was searched starting in 1982 (the beginning of the TME era) until December 2022, using the following search terms: ((((((((((total neoadjuvant therapy) OR (total neoadjuvant treatment)) OR (neoadjuvant treatment)) OR (neoadjuvant chemotherapy)) OR (preoperative chemotherapy)) OR (preoperative chemoradiotherapy)) OR (preoperative chemoradiation)) OR (neoadjuvant chemoradiotherapy)) OR (neoadjuvant chemoradiation)) AND (rectal cancer)) AND (randomised trial). Citation tracking of the included articles was also performed. Only English-language studies were included.

### 2.2. Inclusion and Exclusion Criteria

The study inclusion criteria were based on the following PICO (population, intervention, control, and outcomes) model. Population: patients treated within a frame of randomised study with histologically confirmed adenocarcinoma of the rectum. Intervention: 1. TNT, defined as neoadjuvant radio(chemo)therapy followed by or preceded by neoadjuvant chemotherapy; or 2. neoadjuvant fluoropyrimidine-based chemoradiotherapy with the addition of a platinum derivative or other combined regimen (e.g., with irinotecan). Control: neoadjuvant chemoradiotherapy or neoadjuvant radiotherapy alone. Outcomes: randomised trials providing the following data will be included: 1. OS and DFS or disease-related treatment failure, or QoL in the trials on TNT; 2. data on post-metastases survival for the trials on TNT or other neoadjuvant schedules.

Exclusion criteria were as follows: non-randomised trials; trials assessing treatment regimens that included targeted therapy or biotherapy; trials assessing timing of surgery after neoadjuvant radio(chemo)therapy if the neoadjuvant schedule was the same in both arms or neoadjuvant radiotherapy alone (without neoadjuvant chemotherapy) was used in both arms; trials comparing preoperative vs. postoperative radio(chemo)therapy; trials on recurrent rectal cancer; pathological diagnoses other than adenocarcinoma; planned local excision; and planned w&w.

### 2.3. Study Selection

Articles were screened according to their titles and abstracts. Full texts of all pertinent studies were obtained. In case of multiple publications on the same clinical trial, data with the longest follow-up were used for each outcome.

### 2.4. Data Extraction

OS and QoL were the endpoints for evaluation of the trials on TNT. For the trials on TNT or other neoadjuvant schedules, post-metastatic survival was the endpoint for evaluation. All potentially eligible studies were independently subjected to a full-text review by two authors to assess their suitability according to the predefined inclusion criteria. For each trial, the following data were extracted: total numbers of patients; length of follow-up; the trial’s basic characteristics; and main outcomes. Any disagreements were resolved by discussion. Data extraction was carried out using a predesigned form.

### 2.5. Statistical Methods

The study protocol was written before the literature review. We tested two hypotheses: (1) that TNT results in long-term OS or QoL improvement; and (2) that the ATRESS phenomenon is the reason for the discrepancy between the reduction of distant metastases and the lack of OS improvement with TNT or with other neoadjuvant treatment schedules. The outcome measures were the hazard ratios (HRs) for the post-metastatic survival with the corresponding 95% confidence intervals (CIs) and *p* value.

## 3. Results

### 3.1. Overview of Randomised Trials on TNT

Our literature search identified six randomised studies comparing TNT with neoadjuvant (chemo)radiation for which the data on survival were available (Table 1, Figure 1) [3,4,5,8,12,13,14,15]. Only the Polish II trial by Cisel et al. [12] and GCR-3 trial by Fernandez-Martos et al. [3] had a follow-up longer than 5 years, which precluded quantitative pooled meta-analysis of the long-term OS [6]. OS benefit after TNT was shown only in the STELLAR trial by Jin et al. [15] after the median follow-up of only 35 months. Surprisingly, this improvement was not associated with DFS benefit. The cause of this finding is unclear. The improvement in OS at 3 years without DSF benefit was also seen in the similarly designed Polish II study [13]. However, in this study, updated results after the median follow-up of 7 years showed that the initial OS improvement vanished later resulting in identical survival at 8 years [12]. Therefore, longer follow-up in the STELLAR trial is needed. An improvement in DFS (or in disease-related treatment failure) with TNT was found only in the RAPIDO and the PRODIGE-23 trials [4,5]; however, without corresponding significant OS benefit after a follow-up shorter than 5 years. In the PRODIGE-23 study, an insignificant trend towards better OS in the TNT group was observed, *p* = 0.077 [5]. In contrast, in the RAPIDO trial, actuarial curves showing OS in the TNT cohort and in the chemoradiation cohort overlapped and there was no sign of curves diverging at 5 years [4]. Surprisingly, the RAPIDO trial showed worse local control in the TNT arm compared to the chemoradiation arm (Table 1) [8]. In other trials, there was no difference in local control between randomised arms.

### 3.2. Evaluation of QoL after TNT Given without Considering w&w 

Of the six randomised trials on TNT, the analysis evaluating QoL was presented only in the RAPIDO trial [16]. For the cancer-free patients, apart from neurotoxicity, there were no differences in QoL and in anorectal functions between the chemoradiation group and the TNT group. Three years after surgery, the EORTC QLQ-CIPN20 questionnaires’ analysis revealed statistically significant and clinically meaningful differences with worse scores for TNT compared with the chemoradiation group for the sensory scale (*p* < 0.0001). When the TNT group was compared with the patients from the chemoradiation group who did not receive postoperative chemotherapy, the differences were statistically and clinically significant for sensory, motor, and overall QLQ-CIPN20 scores; three years after surgery, any grade neurotoxicity affected 11% more patients from the TNT group than from the chemoradiation group; 14% vs. 3%, *p* = 0.002. The above analysis did not evaluate QoL during neoadjuvant treatment and was performed after excluding the patients suffering from local or distant failure. The toxicity during neoadjuvant treatment was higher in the TNT group compared with the chemoradiation group; the rate of patients with grade 3+ toxicity was 48% vs. 25%, respectively [4]. Moreover, neoadjuvant treatment lasted 13 weeks longer in the TNT group compared with the chemoradiation group; 19 weeks (1 week of short-course radiation plus 18 weeks of neoadjuvant chemotherapy) vs. 6 weeks of chemoradiation, respectively. The local failure affected 3% more patients in the TNT group than in the chemoradiation group (Table 1) [8]. All the aforementioned multifactorial impairment of QoL after TNT should be weighed against the QoL benefit of TNT resulting from an absolute 8% reduction in the incidence of distant metastases.

### 3.3. Evaluation of ATRESS after TNT

Of the six randomised trials on TNT, the ATRESS phenomenon was identified in the RAPIDO trial [8]. Median post-metastatic survival was 0.7 year shorter in the TNT group than in the chemoradiation group; 2.4 years (interquartile range 0.8–4.1) vs. 3.1 years (interquartile range 1.4–6.6), respectively, HR = 1.40 (95% CI 1.01–1.94), *p* = 0.04 [8]. In the Polish II study co-authored by the present authors, the database was available for relevant analysis. In this analysis, ATRESS was not identified. Median post-metastatic survival in the TNT group compared with the control group was 20 months (95% CI 13–27) vs. 12 months (95% CI 8–16), respectively, log-rank *p* = 0.30, HR = 0.84 (95% CI 0.61–1.17). In the remaining four trials on TNT, there were no published data that would allow the evaluation of whether ATRESS impacted OS. Quantitative pooled meta-analysis of the presence of ATRESS was not performed because of heterogeneity between trials in the duration of neoadjuvant chemotherapy: 18 weeks in the RAPIDO trial and only 6 weeks in the Polish II study.

### 3.4. ATRESS in Other Trials Exploring Neoadjuvant Therapy

Our literature review identified 29 trials exploring neoadjuvant therapy in rectal cancer (Figure 1). Of these, only in the PETACC 6 trial relevant data were published [17]. The authors of this trial tried to find out the cause explaining why, in the experimental arm, adding oxaliplatin to pre- and postoperative capecitabine resulted in numerically fewer patients with distant metastases, but more patients died compared with the control arm without oxaliplatin. The analysis of post-metastatic survival showed the insignificant trend towards worse survival in the oxaliplatin arm compared with the arm without oxaliplatin (HR = 1.27 95% CI 0.90–1.80), *p* = 0.18, and the authors hypothesised that ATRESS, along with the higher rate of non-cancer deaths in the oxaliplatin arm, explains why the reduction of distant metastases did not translate to the OS improvement [17].

## 4. Discussion

Our review shows that the follow-ups of the published TNT studies are too short yet to resolve whether TNT improves long-term OS, and suggests that QoL might not have been improved with TNT if all patients are being considered for radical resection. Therefore, currently, in the light of unproven OS benefit, the gain of TNT (reduction in distant metastasis) does not seem to outweigh the harm from excess acute and late toxicity, and longer duration of neoadjuvant treatment. For this reason, the utility of TNT can be questioned; i.e., routine use of TNT is not sufficiently justified by the current knowledge. This statement is valid for institutions where adjuvant chemotherapy is not the standard of care, but might not be valid for institutions where adjuvant chemotherapy is adopted. This problem is highlighted by the RAPIDO trial where approximately half of the patients from the chemoradiation group did not receive adjuvant chemotherapy because of institutional guidelines [4]. These patients benefited from avoiding toxicity of the (neo)adjuvant chemotherapy, whereas both the remaining patients from the chemoradiation group and the patients from the TNT group experienced toxicity of the (neo)adjuvant chemotherapy. The policy of avoiding adjuvant chemotherapy after neoadjuvant radio(chemo)therapy is based on meta-analyses that did not show a benefit in OS after adjuvant chemotherapy compared with observation [1,2]. Accepting that adoption of toxic experimental treatment is only justified when improvement in long-term OS or in QoL is shown, neither delivery of TNT nor adjuvant chemotherapy is evidence-based.

The comparison of the six evaluated studies was hampered because of the different study protocols, different lengths of the follow-ups, and different disease stages included. The RAPIDO trial suggests that the ATRESS can explain, at least in part, the absence of OS benefit after TNT. However, the analysis performed in the database of the Polish II study did not show the presence of ATRESS. The patient populations differed between these two trials, precluding the direct comparison. The initial disease stage was more advanced in the Polish study, in which 63% of patients had T4 disease [13], compared with 31% in the RAPIDO trial [4]. In accordance with this difference, patients in the Polish trial had a much shorter post-metastatic survival than in the RAPIDO trial. However, the reason why the post-metastatic survival of the Polish TNT group was much longer than that of the control group is unclear. It may be explained by the difference in duration of neoadjuvant chemotherapy; 18 weeks in the RAPIDO trial and only 6 weeks in the Polish II trial. This comparison demonstrates that many factors influence the effect of (neo)adjuvant therapies on the course of rectal cancer and the duration of post-metastatic survival. Certainly, the ATRESS deserves more attention as the reports of most randomised trials have not provided relevant information. The current data are insufficient to evaluate the role of ATRESS both in TNT and in other neoadjuvant schedules. To further elucidate the role of ATRESS, post-relapse survival should be compared between arms of randomised trials exploring neoadjuvant treatment in which benefit in DFS does not translate to OS benefit. Documentation and ideally standardization of post-metastatic therapy would also be desirable.

Excess of non-cancer deaths resulting from toxicity of TNT could be potentially another reason explaining the discrepancy between a DFS improvement and an absence of OS improvement. However, the data did not confirm this hypothesis because in the RAPIDO trial, the rate of non-cancer death was 2.4% (11 of 462) in the TNT group vs. 1.8% (8 of 450) in the chemoradiation group, and the corresponding numbers in the PRODIGE-23 trials were 1.7% (4 of 231) vs. 2.6% (6 of 230) [4,5].

The ATRESS acronym was proposed by Fink [9]. This phenomenon was initially observed in previous randomised trials testing FU-based adjuvant chemotherapy against observation in colon cancers [18,19]. The MOSAIC and NSABP C-07 randomised studies testing oxaliplatin-based adjuvant chemotherapy against FU-only adjuvant chemotherapy in stages II and III colon cancer showed reduced post-metastatic survival after oxaliplatin [20,21]. In contrast, the pooled analysis of individual patients’ data from four other randomised trials comparing FU adjuvant chemotherapy against oxaliplatin-based adjuvant chemotherapy in stage III colon cancer did not confirm this observation [22]. However, of four trials included in this analysis, only in one trial was FU adjuvant chemotherapy directly compared in the randomised setting against oxaliplatin-based adjuvant chemotherapy. This could be a reason for the discrepancy between this analysis and the findings from the MOSAIC and the NSABP C-07 trial. Recently, the ATRESS was shown in the JCOG0603 trial comparing hepatectomy followed by chemotherapy vs. hepatectomy alone for liver-only metastatic colorectal cancer [23].

There are several possible mechanisms explaining ATRESS [9]. First, neoadjuvant chemotherapy may kill chemosensitive subclinical metastases or induce chemoresistance in the remaining tumour clonogens. Thus, residual disease could be chemoresistant and more aggressive as suggested by mass-spectroscopy genotyping for somatic gene mutations in colorectal liver metastases, which showed that previous treatment with oxaliplatin-based adjuvant therapy may provide a selection pressure favouring a chemotherapy-resistant more aggressive form of metastatic diseases [24]. Second, neoadjuvant chemotherapy is actually the “first line” treatment, and chemotherapy for overt metastases is the “second line”, i.e., less effective. Third, fewer patients may undergo salvage surgery because of metastases in the (neo)adjuvant chemotherapy group compared with the control group without chemotherapy or with less aggressive chemotherapy [25]. Fourth, oxaliplatin given in the (neo)adjuvant setting may restrict its use in the patients with relapse [25]. Finally, previous chemotherapy reduced the effect of stereotactic radiotherapy of lung and liver metastases [26,27].

## 5. Conclusions

Our review suggests that QoL might not have been improved with TNT if all patients are being considered for radical resection. Therefore, in the light of unproven OS benefit, the gain (reduction of distant metastasis) of TNT does not seem to outweigh the harm (excess of toxicity). (Neo)adjuvant therapy influences the course of rectal cancer for a much longer time than most oncologists are aware. To elucidate the role of ATRESS as a possible cause of an absence of OS improvement despite DFS benefit, post-relapse survival should be compared between arms of randomised trials exploring neoadjuvant treatments.

## Figures and Tables

**Figure 1 cancers-15-01016-f001:**
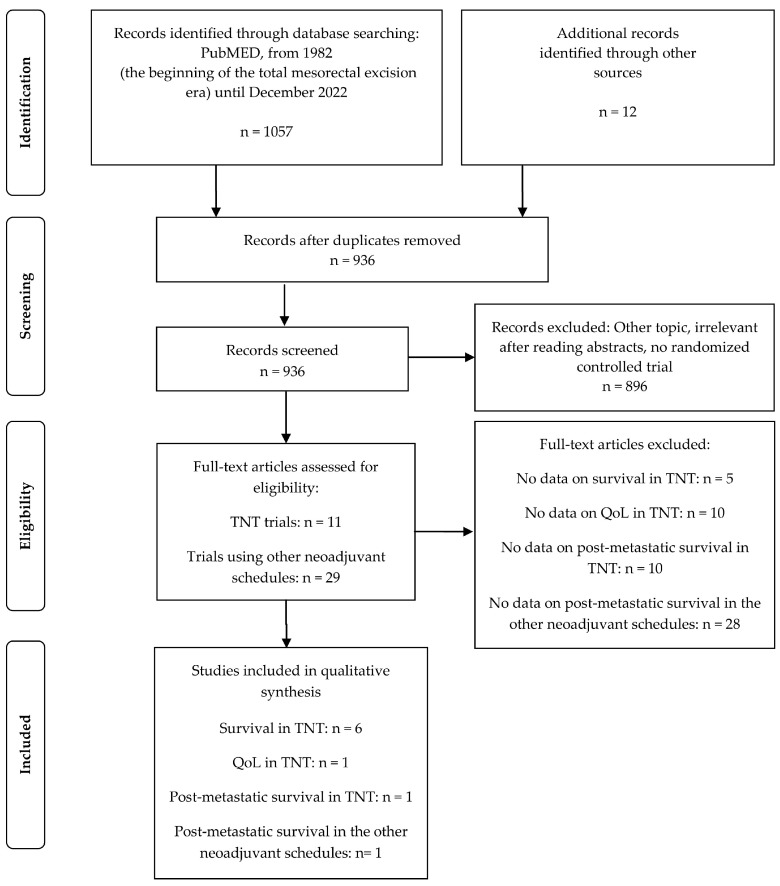
Flow diagram of study retrieval for the systematic review.

**Table 1 cancers-15-01016-t001:** Randomised trials on rectal cancer comparing total neoadjuvant treatment (TNT) against neoadjuvant (chemo)radiation (control).

Study	Number of Patients	Design				Median Follow-Up (Years)	DFS Results (TNT vs. Control)	OS Results (TNT vs. Control)	LRF Results (TNT vs. Control)	Survival After the Onset of DM in Years (TNT vs. Control)	Remarks
		TNT	Neoadjuvant (chemo)Radiation	Adjuvant Chemotherapy			DFS (%), *p*-Value	OS (%), *p*-Value	LRF (%), *p*-Value		
				TNT	Control						
GCR-3 [3] 2015	108	CAPOX (12 weeks) + CRT: 50.4 Gy + Cape and Ox	CRT: 50.4 Gy + Cape and Ox	No postoperative chemotherapy	CAPOX 4x	5.8	5-year: 62 vs. 64, *p* = 0.85, HR not given	5-year: 75 vs. 78, *p* = 0.64, HR not given	5-year: 5 vs. 2, *p* = 0.61, HR not given	No data	Pure design; only the sequence of chemotherapy differed (neoadjuvant vs. adjuvant).
POLISH II [12,13] 2019	515	RT 5x5Gy + FOLFOX or FU+LV (6 weeks)	50.4 Gy + 5-FU and LV or 5-FU, LV and OX	^c^ Optional	^c^ Optional	7.0	8-year: 43 vs. 41, *p* = 0.65, HR 0.95 (95% CI 0.75–1.19) 3-year: 53 vs. 52, *p* = 0.85, HR 0.96 (95% CI 0.75–1.24)	8-year: 49 vs. 49, *p* = 0.38, HR 0.90 (95% CI 0.70–1.15) 3-year: 73 vs. 65, *p* = 0.046, HR 0.73 (95% CI 0.53–1.01)	8-year: 35 vs. 32, *p* = 0.60, HR 1.08 (95% CI 0.70–1.23)	Median 1.7 (95% CI 1.1–2.2) vs. 1.0 (95% CI 0.7–1.4), HR 0.84 (95% CI 0.61–1.17), *p* = 0.30.	Short-course radiation was used in the TNT group and chemoradiation in the control group. Neoadjuvant chemotherapy was used only for 6 weeks.
KIR [14] 2021	180 ^a^	FOLFOX (12 weeks) + HDRBT	HDRBT	FOLFOX (12 weeks)	FOLFOX (24 weeks)	4.0	5-year: 72.3 vs. 68.3, *p* = 0.74, HR not given	5-year: 83.8 vs. 82.2, *p* = 0.53, HR not given	5-year: 6.3 vs. 5.8, *p* = 0.71, HR not given	No data	The use of HDRBT, which is not a widely accepted standard.
RAPIDO [4,8] 2021	912	RT 5x5Gy + CAPOX or FOLFOX (18 weeks)	50.4 Gy + Cape	No postoperative chemotherapy	^d^ Optional; no postoperative chemotherapy or CAPOX/FOLFOX (24 weeks)	4.6	^b^ 3-year: 23.7 vs. 30.4, *p* = 0.019, HR 0.75 (95% CI 0.60–0.93)	3-year: 89.1 vs. 88.8, *p* = 0.59, HR 0.92 (95% CI 0.67–1.25)	5-year: 10 vs. 7, *p* = 0.038, HR 1.60 (95% CI 1.02–2.49)	2.4 (IQR 0.8–4.1) vs. 3.1 (IQR 1.4–6.6) years, HR 1.40 (95% CI 1.01–1.94), *p* = 0.04	Short-course radiation was used in the TNT group and chemoradiation in the control group.
PRODIGE 23 [5] 2021	461	FOLFIRINOX (12 weeks) + CRT: 50.4 Gy+Cape	50.4 Gy + Cape	mFOLFOX or Cape (12 weeks)	mFOLFOX 12x or Cape 8x	3.9	3-year: 75.7 vs. 68.5, *p* = 0.034, HR 0.69 (95%CI 0.49–0.97)	3-year: 90.8 vs. 87.7, *p* = 0.077, HR 0.65 (95% CI 0.40–1.05)	3-year: 4.3 vs. 5.7, *p* = 0.51, HR 0.65, 95% CI 0.40–1.05	No data	Irinotecan was used only in the TNT group.
STELLAR [15] 2022	599	RT 5x5 Gy + CAPOX (12 weeks)	50 Gy + Cape	CAPOX (6 weeks)	CAPOX (18 weeks)	2.9	3-year: 64.5 vs. 62.3, *p* = 0.883, HR 0.883 (95% CI NA–1.11)	3-year: 86.5 vs. 75.1, *p* = 0.033, HR 0.67, (95% CI 0.46–0.97)	3-year: 8.4 vs. 11, *p* = 0.46, HR 0.80 (95% CI 0.45–1.44)	No data	

^a^ randomisation 2:1 to TNT or neoadjuvant HDRBT alone; ^b^ DRTF—disease-related treatment failure ^c^ Delivering postoperative chemotherapy and its schedule was left to the discretion of treating physicians; 39% of patients in both arms eventually received postoperative chemotherapy; ^d^ Predefined by hospital policy; 42% of patients in the control arm eventually received postoperative chemotherapy. TNT—total neoadjuvant treatment; CRT—chemoradiation; DFS—disease-free survival; OS—overall survival; LRF—cumulative probability of locoregional failure; DM—distant metastases; HR—hazard ratio; CI—confidence interval; IQR—interquartile range; HDRBT—high dose rate brachytherapy; CAPOX—capecitabine and oxaliplatin; FOLFOX—fluorouracil, leucovorin and oxaliplatin; FOLFIRINOX—fluorouracil, leucovorin, irinotecan and oxaliplatin; mFOLFOX—modified FOLFOX; Cape—capecitabine; Ox—oxaliplatin; FU—fluorouracil; LV—leucovorin, NA—not applicable.

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
