# Peer review of "Does the Gain of Total Neoadjuvant Therapy Outweigh the Harm in Rectal Cancer? Importance of the ATRESS (neoAdjuvant Therapy-RElated Shortening of Survival) Phenomenon: A Systematic Review"

_cancers, 2023, doi:10.3390/cancers15041016_

Round 1

Reviewer 1 Report

The authors reviewed six randomized studies to investigate whether the benefits of total neoadjuvant therapy (TNT) for rectal cancer outweighs the harm of this intensive treatment

The authors reviewed six randomized trials to investigate whether the benefits of total neoadjuvant therapy (TNT) for rectal cancer outweighs the harms of this intensive treatment. They exclude the sometimes possible omission of surgery after TNT, because this is not yet generally established.
The authors also introduce the acronym ATRESS (neoAdjuvant Therapy-Related Shortening of Survival) into the discussion of TNT and state already in the abstract that ATRESS refers exclusively to the shortening of postmetastatic survival; this serves to avoid misunderstandings. Perhaps for the same reason, consideration should be given to adding "after neoadjuvant chemoradiation" to the rejection of adjuvant chemotherapy for rectal cancer, for example, on page 2, line 63.
In Table 1 of the paper (page 4 below), data from the RAPIDO study have obviously been confused: in the second line concerning this study, the numbers „23.7 vs. 30.4" are given in the column "DFS results". According to the original RAPIDO article (citation 4 of the reviewed paper), however, these are the data for "disease-related treatment failure".
The Comparison of the six studies evaluated is difficult because different study protocols were used, the follow-up was of different lengths, and the patients had different disease stages. For example, 63% of patients in the Polish study (Cisel 2019) had stage T4 tumors, but only 31% of patients in the RAPIDO study. A significant ATRESS effect, i.e. a reduction in postmetastatic survival in the more intensively treated TNT group, was found only in the RAPIDO trial with 2.4 years in the TNT group and 3.1 years in the control group.  In the Polish study, in accordance with the more advanced initial stage, patients had a much shorter postmetastatic survival compared with the RAPIDO study. But surprisingly the postmetastatic survival of the Polish TNT group was thereby with 1.7 years much longer than that of the control group with 1.0 years; the reason for this is unclear.
This comparison demonstrates, that many factors influence the effect of (neo)adjuvant therapies on the course of rectal cancer and the duration of postmetastatic survival. Among these factors, documentation and ideally standardization of postmetastatic therapy would also be desirable.

     The present work is a very good basis for such studies, and it demonstrates once again that adjuvant therapy influences the course of a tumor for a much longer time than most oncologists are aware.

Reviewer 2 Report

This systematic review was conducted with clear and validated methods. Its purpose is to pool data from randomized, prospective trials, to revisit a few oncologic and quality of life endpoints in a patient population affected by rectal cancer and considering total neoadjuvant therapy. The findings and conclusions of the author are presented clearly and add value to the existing literature on this topic.

A few minor stylistic suggestions:

-page 2, line 71: "fit. It is also important to answer the question of what are the reasons for which the reduction of distant metastases did not translate into OS improvement with TNT". This sentence needs to be rephrased for clarification. Consider "It is also important to assess the reasons for which the reduction of distant metastases did not translate into OS improvement with TNT"

-in various portions of the manuscript, "if non-operative management is not considered" is a bit of a complicated choice of words, when thinking of a broader readership. Please consider a simpler phrase, such as "if all patients are being considered for radical resection", or something of the sort.
